

# Evaluation of 17 microsatellite markers for parentage testing and individual identification of domestic yak (*Bos grunniens*)

Jie Pei[1,2], Pengjia Bao[1,2], Min Chu[1,2], Chunnian Liang[1,2], Xuezhi Ding[1,2], Hongbo Wang[1,2], Xiaoyun Wu[1,2], Xian Guo[1,2] and Ping Yan[1,2]

[1] Animal Science Department, Lanzhou Institute of Husbandry and Pharmaceutical Sciences, Chinese Academy of Agricultural Sciences, Lanzhou, Gansu, China
[2] Key Laboratory for Yak Genetics, Breeding, and Reproduction Engineering of Gansu Province, Lanzhou, Gansu, China

## ABSTRACT

**Background:** Yak (*Bos grunniens*) is the most important domestic animal for people living at high altitudes. Yak ordinarily feed by grazing, and this behavior impacts the accuracy of the pedigree record because it is difficult to control mating in grazing yak. This study aimed to evaluate the pedigree system and individual identification in polled yak.

**Methods:** A total of 71 microsatellite loci were selected from the literature, mostly from the studies on cattle. A total of 35 microsatellite loci generated excellent PCR results and were evaluated for the parentage testing and individual identification of 236 unrelated polled yaks. A total of 17 of these 35 microsatellite loci had polymorphic information content (PIC) values greater than 0.5, and these loci were in Hardy–Weinberg equilibrium without linkage disequilibrium.

**Results:** Using multiplex PCR, capillary electrophoresis, and genotyping, very high exclusion probabilities were obtained for the combined core set of 17 loci. The exclusion probability (PE) for one candidate parent when the genotype of the other parent is not known was 0.99718116. PE for one candidate parent when the genotype of the other parent is known was 0.99997381. PE for a known candidate parent pair was 0.99999998. The combined PEI (PE for identity of two unrelated individuals) and PESI (PE for identity of two siblings) were >0.99999999 and 0.99999899, respectively. These findings indicated that the combination of 17 microsatellite markers could be useful for efficient and reliable parentage testing and individual identification in polled yak.

**Discussion:** Many microsatellite loci have been investigated for cattle paternity testing. Nevertheless, these loci cannot be directly applied to yak identification because the two bovid species have different genomic sequences and organization. A total of 17 loci were selected from 71 microsatellite loci based on efficient amplification, unambiguous genotyping, and high PIC values for polled yaks, and were suitable for parentage analysis in polled yak populations.

Corresponding authors
Xian Guo, guoxian@caas.cn
Ping Yan, pingyan63@126.com

## INTRODUCTION

Yak (*Bos grunniens*), a member of Bovidae family, has successfully been adapted to the severe cold and low oxygen levels, characteristics of high altitude regions (~2,500–5,500 m), such as the Himalayas in South-Central Asia, the Qinghai-Tibetan Plateau, Mongolia, and Russia (*Wu, 2016*). These regions are known for their high elevations, pristine natural environments, and extreme seasonal variations (*Mizuno et al., 2015*). The ability of yak to survive in such rugged natural environment is due to its varied behavioral, physiological, and genetic adaptations (*Barsila et al., 2014*; *Hu et al., 2012*; *Qiu et al., 2012*). For instance, remarkable reduction of heat production at night when not grazing and increased energy consumption when grazing under free-range conditions enables yaks to save more energy and resist the extremely harsh conditions than other cattle under similar environmental conditions (*Ding et al., 2014*). Yak can thrive under extreme environmental conditions, such as the Tibetan Plateau where few other animals can survive. In this region, yak have significantly contributed to human life by providing meat, milk, fur, leather, and transportation compared to other animals (*Medhammar et al., 2012*; *Wang et al., 2018*).

There are about 13 million domestic yak in China, accounting for approximately 90% of the global yak population. Although there are 18 yak breeds in China, only one breed (Datong) has been generated in a breeding program (*Wu, 2016*). However, polled yak have been bred for many years at the foot of Ashidan Mountain, as polling reduces the risk of horn-inflicted injury or death among the herdsmen. Accurate genealogical records can help estimate genetic parameters and improve breeding programs, ensuring efficient and effective breeding progression to avoid excessive inbreeding. However, it is difficult to control mating among yaks while grazing. Furthermore, semen samples could be erroneously mislabeled during preparation, and mating records might be misinterpreted because of clerical errors made during artificial insemination. Therefore, accurate yak pedigree records that are compiled by parentage testing and individual identification are essential to the yak breeding process.

Molecular markers can indicate the degree of genetic relatedness between animals, facilitating parentage verification and individual identification (*Estoup, Jarne & Cornuet, 2002*). Microsatellites refer to short tandem repeats (STRs) or simple sequence repeats (SSRs), and are considered as tracts of DNA motifs ranging from one to 10 nucleotides in length with repeats of 5–50 times (*Carneiro Vieira et al., 2016*). Microsatellites can be used to develop pedigree animal populations and evaluate animal breeding, supporting genetic improvement by selective breeding (*Weising et al., 1997*). The application of microsatellites as molecular markers for animal identification and parentage verification produced highly accurate and effective results in both breeding and forensics (*Linacre et al., 2011*).

Microsatellite marker analysis has been used to verify the parentage in breeding registries and identify individual animals that are linked to a particular database or owner. Microsatellite panels of cattle (*Zhao et al., 2017*), horse (*Kang et al., 2016*), sheep (*Rosa et al., 2013*), dog (*Jeong et al., 2015*), and parrot (*Coetzer et al., 2017*) have been well characterized. Parentage control in the beef cattle breeds, Charolais, Limousin, and

Preta, in Portugal was assessed using 10 microsatellite markers, and the results revealed a combined exclusion probability (PE) above 0.9995, indicating their ability to exclude a random parent pair (*Carolino et al., 2009*). The application of 11 microsatellite loci in paternity testing in Yugoslav Pied cattle breed in Serbia revealed a combined PE of 0.999 (*Stevanovic et al., 2010*). A total of 16 specific microsatellite markers were used to develop a genetic system of meat traceability for several beef cattle breeds, including Japanese Black, Anduo yak, Limousin, Jiaxian Red, Nanyang Yellow, and Luxi Yellow (*Zhao et al., 2017*). Previous studies reported that microsatellite genotyping was used for population genetics analysis and parentage testing in yak. However, these loci and their primers were originally developed for cattle, and then used directly in yak (*Li et al., 2013*; *Nguyen et al., 2005*). Therefore, there is a need for explorative and application-based studies on microsatellite markers or panels that are suitable for accurate individual identification and parentage testing in yak.

Hence, in the present study, we aimed to establish a paternity test and individual identification system for the polled yak. Therefore, the study was expected to (1) calculate the genetic parameters of polled yak microsatellite loci, which have been commonly reported by population genetic studies of cattle and yak; (2) evaluate the application values of the loci with high polymorphic information content (PIC) for parentage testing and individual identification; and (3) explore a multi-loci combination test system for parentage testing and individual identification.

## MATERIALS AND METHODS

### Marker selection and primer design

The microsatellites used in the present study were selected from the previous reports on cattle breeding based on the following criteria: (a) high PIC; (b) large number of alleles; (c) relatively infrequent null alleles; and (d) homogeneous or approximately homogeneous repeat motifs (*Schnabel, Ward & Derr, 2000*). Among the 71 bovine microsatellite markers selected, 65 were derived from cattle, and six were exclusive to yak (Table S1). All 71 microsatellites and their flanking sequences were found in the cattle genome, and were searched for in the yak genome. The primers for most loci used in previous studies were not suitable for the yak genome because of low scores. However, the primers used to amplify BM1824, BM2113, BMS2533, ETH121, ETH225, ILSTS008, INRA124, RM099, INRA126, UMN0103, UMN0307, UMN0920, UMN2303, UMN3007, and UMN3008 loci presented relatively high scores, so these loci were not redesigned for yak. The primers used in the present study are listed in Table S1.

### Sample collection

Polled yaks bred via selective breeding were selected from herds in Ashidan Mountain region in Qinghai Province, China. To avoid consanguinity, samples were taken from animals with no genetic relationship. All yaks were handled in strict accordance with good animal practices by following the *Animal Ethics Procedures and Guidelines of the People's Republic of China*. The present study was approved by the *Animal Administration and Ethics Committee of Lanzhou Institute of Husbandry and Pharmaceutical Sciences*

*of Chinese Academy of Agricultural Sciences* (Permit No. SYXK-2016-0039). Blood was drawn from the jugular veins of 236 unrelated individuals, including 38 sires and 198 dams, and samples were mixed with preservation buffer (containing 1.5 mg mL$^{-1}$ EDTA and 137 mmol L$^{-1}$ NaCl) at a ratio of 5:1. The blood samples were stored at $-80$ °C in an ultra-cold freezer until DNA extraction.

## DNA extraction and quantification

Genomic DNA was extracted from white blood cells, separated from whole blood, and digested with proteinase K. After digestion, the samples were centrifuged at $5,000 \times g$ for 2 min, and the resulting supernatant (clear aqueous layer) was transferred to a new test tube. After the addition of 0.5 mL of 10 mg mL$^{-1}$ RNase A, DNA was individually extracted with a phenol:chloroform:isoamyl alcohol mixture (25:24:1) followed by chloroform, precipitated with ethanol, and resuspended in 50 mL TE buffer (10 mM Tris–HCl and 1 mM EDTA, pH 8.0). The extracted DNA was then quantified using a NanoDrop 2000 fluorometer (Thermo Fisher Scientific, Waltham, MA, USA).

## Preliminary primer screening

Unlabeled primer pairs (Table S1) for each microsatellite marker were used for amplification of DNA fragments. PCR was performed with a reaction mixture at a total volume of 20 μL, comprising of 20–50 ng genomic DNA, 10 mM Tris–HCl (pH 9.0), 50 mM KCl, 1.5 mM MgCl$_2$, 2.0 mM each dNTP, five ng of bovine serum albumin, and 1.0 U of *Taq*™ Hot Start Version polymerase (TaKaRa Bio Inc., Kusatsu, Shiga, Japan). Primer concentrations ranged from 1.0 to 5.5 μM. The thermal cycle parameters were as follows: 5 min at 95 °C; 35 cycles of 30 s at 95 °C, 30 s at 55–58 °C (Table S1), and 20 s at 72 °C; and a final extension step for 5 min at 72 °C. Amplifications were performed on a Veriti 96-Well Thermal Cycler (Applied Biosystems Corp., Foster City, CA, USA). The amplicons were visualized on 2% agarose gels (Gene Tech Co. Ltd, Chai Wan, Hong Kong, People's Republic of China). Five μL of PCR product were mixed with one μL of loading dye (TaKaRa Bio Inc., Kusatsu, Shiga, Japan) and electrophoresed for 35 min at 85 V. The samples were then visualized using the GelDoc™ XR+ gel imaging system (Bio-Rad Corp., Hercules, CA, USA).

The amplicons with high specificity and high amplification efficiency, whose loci were either non-syntenic or separated by >10 cM (to avoid strong genetic linkage) (Table S2), were manually cut out of agarose gels and forwarded to Invitrogen (Carlsbad, CA, USA) and Thermo Fisher Scientific without prior purification for Sanger sequencing on an ABI3730xl automated sequencer (Applied Biosystems Corp., Foster City, CA, USA) to identify repeat markers. Only the forward primers were used to sequence the target markers.

## Genetic information acquisition

The confirmed primer sequences flanking the microsatellite loci were synthesized with a fluorescent label (FAM™, HEX™, or TAMRA™; Thermo Fisher Scientific, Waltham, MA, USA) attached to the 5′ end of each forward primer. The microsatellites were

separately amplified by PCR to identify the highly polymorphic loci. The amplification systems and conditions were similar to those described above for unlabeled primers.

After amplification, one µL of amplified fragment mix was added to 0.5 µL of loading buffer (blue dextran, 50 mg mL$^{-1}$; EDTA, 25 mM) and four µL of deionized formamide, and it was then denatured by incubation for 5 min at 95 °C. An internal size standard (0.5 µL of Thermo ABI 4322682; Thermo Fisher Scientific, Waltham, MA, USA) was added to each sample, and the fluorescently labeled PCR products were then separated by capillary electrophoresis (ABI3730xl Genetic Analyzer; Applied Biosystems Corp., Foster City, CA, USA). The fluorescently labeled DNA fragments were first analyzed with GENESCAN v. 3.7 (Applied Biosystems Corp., Foster City, CA, USA) followed by GENOTYPER v. 3.7 NT (Applied Biosystems Corp., Foster City, CA, USA). The fragments were then automatically sorted according to the internal size standard. The numbers of microsatellite repeat motifs were calculated based on the amplicon lengths (Table S3).

## Genetic information analysis

The genotypic data were initially processed in Microsoft Office Excel 2007, manually checked for errors, and then transformed into input files that were required for subsequent analyses. The performance characteristics (observed heterozygosity ($H_O$), expected heterozygosity ($H_E$), PIC, estimated null allele frequency (F(null)), Hardy–Weinberg equilibrium (HWE), and linkage disequilibrium) were measured with GENEPOP v. 4.6 (*Raymond & Rousset, 1995*; *Rousset, 2008*) and CERVUS v. 3.0.7 (*Kalinowski, Taper & Marshall, 2010*; *Slate, Marshall & Pemberton, 2000*).

## Multiplex PCR conditions

Genotyping of 236 yaks for 35 loci (Table 1) produced a core set of 17 loci with high PIC values (Table 2). Four multiplex PCR reactions were assembled, each containing four or five microsatellite markers. The primer sequences and concentrations used in the multiplex PCR reactions are shown in Table 2. The multiplex PCR reactions were performed with reaction mixtures at a total volume of 15 µL, containing 25 ng of genomic DNA, 10 mM Tris–HCl (pH 8.3), 35 mM KCl, 1.8 mM MgCl$_2$, 5.0 mM each dNTPs, and 2.5 U of *Taq*™ Hot Start Version polymerase (TaKaRa Bio Inc., Kusatsu, Shiga, Japan). Amplifications were performed in a Veriti 96-Well thermal cycler (Applied Biosystems Corp., Foster City, CA, USA) under the following conditions: 95 °C for 5 min; 25 cycles of 95 °C for 30 s, 55–58 °C for 30 s, and 72 °C for 30 s; 10 cycles of 95 °C for 30 s, 53 °C for 30 s, and 72 °C for 30 s; and final extension at 72 °C for 5 min. Genotyping of these microsatellite loci was performed as described above.

## Parentage testing and individual identification

PE values for parentage testing were obtained using a likelihood-based method based on genotypic information. Five types of PEs were calculated for the set of loci in CERVUS v. 3.0.7. PE1 was defined as the average probability of excluding an unrelated candidate parent of an arbitrary offspring when the genotype of the other parent is unknown. PE2 was defined as the average probability of excluding an unrelated candidate parent

**Table 1 Genetic information for the 35 polymorphic microsatellite loci.**

| Locus ID | Range (bp) | Repeat motif | $N_A$ | $H_O$ | $H_E$ | PIC | HWE | F(Null) |
|---|---|---|---|---|---|---|---|---|
| **BM720** | 160–168 | AC | 5 | 0.574 | 0.588 | 0.541 | NS | 0.0127 |
| BM1818 | 239–251 | AC | 5 | 0.325 | 0.329 | 0.302 | NS | 0.0031 |
| **BM2113** | 115–133 | TG | 8 | 0.673 | 0.662 | 0.623 | NS | 0.0067 |
| BM2943 | 253 | AC | 1 | 0 | 0 | 0 | ND | ND |
| CSSM013 | 156–160 | TG | 3 | 0.294 | 0.309 | 0.272 | NS | 0.0180 |
| CSSM029 | 182–194 | AC | 6 | 0.517 | 0.466 | 0.417 | NS | 0.0623 |
| CSSM033 | 309–317 | TG | 3 | 0.137 | 0.130 | 0.126 | ND | 0.0261 |
| **CSSM036** | 157–179 | TG | 10 | 0.677 | 0.693 | 0.646 | NS | 0.0116 |
| CSSM041 | 128–134 | TG | 4 | 0.457 | 0.428 | 0.402 | NS | 0.0427 |
| CSSME070 | 249–259 | AC | 5 | 0.513 | 0.526 | 0.412 | NS | 0.0093 |
| **HAUT24** | 217–231 | AC | 8 | 0.628 | 0.633 | 0.572 | NS | 0.0059 |
| HEL5 | 214–228 | TG | 6 | 0.182 | 0.181 | 0.174 | ND | 0.0055 |
| **HEL6** | 247–265 | GT | 8 | 0.686 | 0.754 | 0.721 | NS | 0.0427 |
| HEL10 | 171–177 | TG | 4 | 0.091 | 0.502 | 0.417 | *** | 0.7025 |
| ILSTS006 | 162–170 | GT | 4 | 0.453 | 0.448 | 0.374 | NS | 0.0095 |
| ILSTS008 | 172–184 | AC | 6 | 0.577 | 0.550 | 0.446 | NS | 0.0257 |
| **ILSTS028** | 261–293 | GT | 11 | 0.806 | 0.838 | 0.815 | NS | 0.0194 |
| ILSTS030 | 150–156 | GT | 4 | 0.106 | 0.129 | 0.123 | ND | 0.1033 |
| **INRA005** | 186–202 | CA | 5 | 0.607 | 0.604 | 0.529 | NS | 0.0058 |
| **INRA023** | 184–204 | TG | 10 | 0.753 | 0.780 | 0.752 | NS | 0.0162 |
| INRA035 | 104 | TG | 1 | 0 | 0 | 0 | ND | ND |
| **INRA037** | 296–312 | CA | 8 | 0.688 | 0.770 | 0.730 | NS | 0.0576 |
| **INRA063** | 175–187 | TG | 7 | 0.604 | 0.642 | 0.585 | NS | 0.0317 |
| MGTG4B | 245–255 | AC | 5 | 0.196 | 0.273 | 0.254 | ** | 0.1631 |
| MM12 | 109–121 | GT | 4 | 0.085 | 0.087 | 0.084 | ND | 0.0242 |
| **POTCHA** | 128–148 | CA | 8 | 0.573 | 0.585 | 0.540 | NS | 0.0227 |
| RM099 | 233 | CA | 1 | 0 | 0 | 0 | ND | ND |
| **SPS115** | 231–261 | CA | 12 | 0.782 | 0.806 | 0.780 | NS | 0.0146 |
| TGLA57 | 253–263 | GT | 6 | 0.348 | 0.340 | 0.322 | NS | 0.0120 |
| **TGLA126** | 209–223 | TG | 8 | 0.587 | 0.690 | 0.639 | NS | 0.0819 |
| **TGLA227** | 210–222 | AC | 5 | 0.611 | 0.617 | 0.538 | NS | 0.0015 |
| **YAK07** | 323–339 | TG | 9 | 0.63 | 0.621 | 0.589 | NS | 0.0104 |
| **YAK08** | 321–343 | CA | 8 | 0.698 | 0.676 | 0.612 | NS | 0.0218 |
| **YAK11** | 306–314 | GT | 7 | 0.679 | 0.666 | 0.597 | NS | 0.0096 |
| YAK12 | 259–279 | GT | 8 | 0.549 | 0.522 | 0.406 | NS | 0.0269 |

**Notes:**
Font bold, core microsatellite loci are indicated in bold.
Range, range of allele sizes; Repeat motif, repeat motif of microsatellites; $N_A$, number of alleles found; $H_O$, observed heterozygosity; $H_E$, expected heterozygosity; PIC, polymorphism information content; HWE, departure from Hardy–Weinberg equilibrium; NS, not significant; ND, not done; F(Null), estimated null allele frequency.
** Significant at the 1% level;
*** Significant at the 0.1% level.

of an arbitrary offspring when the genotype of the other parent is known. PEP was defined as the average probability of excluding a pair of unrelated candidate parents of an arbitrary offspring. PEI was defined as the average probability of differentiating two randomly

**Table 2 Detailed primer information for the core microsatellite loci identified in the present study.**

| Locus name | Forward primer (5′→3′) | Reverse primer (5′→3′) | C [mM] | Ta (°C) | PMP | PML | Fluoro | RM | EP1 | EP2 | EPP | EI | ESI | CL |
|---|---|---|---|---|---|---|---|---|---|---|---|---|---|---|
| BM720 | GAAATCAACAAGTTCCAATCCTG | ATCTCATTCTTGTGTCATGGATGA | 3.5 | 56 | 3 | 2 | 6-FAM | (AC) | 0.189 | 0.353 | 0.532 | 0.783 | 0.489 | 13 |
| BM2113 | GCTGCCTTCTACCAAATACCC | CTTCCTGAGAGAAGCAACACC | 1.9 | 55 | 1 | 1 | 6-FAM | (TG) | 0.261 | 0.439 | 0.636 | 0.847 | 0.542 | 2 |
| CSSM036 | GATAACTCAACCACACGTCT | AAGAAGTACTGGTTGCCAATCG | 2.8 | 56 | 2 | 1 | 6-FAM | (TG) | 0.283 | 0.456 | 0.644 | 0.859 | 0.561 | 27 |
| HAUT24 | CTCTGCCTTTGTCCCTGTCT | CCAAACCCCTACCCACA | 5.7 | 56 | 2 | 1 | TAMARA | (AC) | 0.221 | 0.377 | 0.552 | 0.805 | 0.517 | 22 |
| HEL6 | GGACACGACTGAGCAAGTAACA | GCTTTGGCAGGCAGATACAT | 4.0 | 56 | 1 | 1 | HEX | (GT) | 0.367 | 0.549 | 0.743 | 0.907 | 0.603 | 1 |
| ILSTS028 | AGAAGAGTGTACCTCCTCCCAC | TCCAGATTTTGTACCAGACCAT | 4.6 | 56 | 1 | 1 | TAMARA | (GT) | 0.503 | 0.673 | 0.847 | 0.953 | 0.656 | 11 |
| INRA005 | CTTCAGGCATACCCTACACCA | GGGGAATCTGTGGAGGAGTT | 8.3 | 56 | 4 | 2 | 6-FAM | (CA) | 0.190 | 0.328 | 0.483 | 0.768 | 0.494 | 12 |
| INRA023 | ATTTCCTTCTGACTGGTACTTC | GTGTCCCTCCTCTAATCCCTAA | 3.0 | 55 | 3 | 2 | HEX | (TG) | 0.408 | 0.589 | 0.782 | 0.924 | 0.620 | 3 |
| INRA037 | GCTACAATCCAGACTGAGCACG | GACACGGCTTAGCGACTGAA | 3.1 | 57 | 3 | 2 | HEX | (CA) | 0.370 | 0.549 | 0.731 | 0.909 | 0.611 | 10 |
| INRA063 | AAACCACAGAAATGCTTGGAAG | ATTTGCACAAGCTAAATCTAACAA | 3.6 | 56 | 1 | 1 | TAMARA | (TG) | 0.228 | 0.390 | 0.566 | 0.816 | 0.524 | 18 |
| POTCHA | ATGCCAACTTTTCCCATCACT | GTAAACACAGTTCCCTGGAGAGA | 3.5 | 56 | 2 | 1 | HEX | (CA) | 0.192 | 0.357 | 0.540 | 0.783 | 0.488 | 15 |
| SPS115 | AAAGTGACACAACAGCTTCACC | ACCGAGTGTCCTAGTTTGGC | 4.6 | 55 | 4 | 2 | TAMARA | (CA) | 0.452 | 0.628 | 0.814 | 0.938 | 0.636 | 15 |
| TGLA126 | ATGAGAGAGGCTTCTGGGATG | CTTCACCATTGGACCACGAG | 3.7 | 56 | 4 | 2 | HEX | (TG) | 0.272 | 0.444 | 0.625 | 0.854 | 0.558 | 20 |
| TGLA227 | CAAAGGAGCATAACTTTACAGCA | AGCCTAACCATTGGACAGC | 4.9 | 57 | 3 | 2 | TAMARA | (AC) | 0.200 | 0.337 | 0.494 | 0.775 | 0.501 | 18 |
| YAK07 | TAACAAAGCTGCTGGGAACAT | CGGAGTCACTTTCCTCACCTAT | 2.4 | 56 | 4 | 2 | HEX | (TG) | 0.230 | 0.412 | 0.615 | 0.825 | 0.516 | 1 |
| YAK08 | ACTGGAGTAGGTTGCCCGCTCTCT | CCTGGCTTGGGTCCTG | 3.8 | 56 | 2 | 1 | HEX | (CA) | 0.247 | 0.405 | 0.572 | 0.831 | 0.545 | 6 |
| YAK11 | TCCCCTCACTCCTCATTGGTACCAGT | TGCAGGCAGTTTCTT | 4.4 | 56 | 3 | 2 | TAMARA | (GT) | 0.233 | 0.387 | 0.548 | 0.820 | 0.537 | 1 |
| Combined | | | | | | | | | 0.99718116 | 0.99997381 | 0.99999998 | >0.99999999 | 0.99999899 | |

**Note:**
C, concentration of primers; Ta, annealing temperature; PMP, panel numbers for multiplex PCR; PML, panel numbers for multiplex loading; Fluoro, fluorescent dye; RM, repeat motif of microsatellites; PEI, the average probability that the set of loci will exclude an unrelated candidate parent from parentage of an arbitrary offspring when the genotype of the other parent is unknown; PE2, the average probability that the set of loci will exclude an unrelated candidate parent from parentage of an arbitrary offspring when the genotype of the other parent is known; PEP, the average probability that the set of loci will exclude a pair of unrelated candidate parents from parentage of an arbitrary offspring; PEI, the average probability that the set of loci will differentiate between two randomly selected individuals; PESI, the average probability that the set of loci will differentiate between two randomly selected full siblings; CL, locations of microsatellite loci on cattle chromosomes.

selected individuals. PESI was defined as the average probability of differentiating two randomly selected full siblings (*Kalinowski, Taper & Marshall, 2010*; *Slate, Marshall & Pemberton, 2000*).

## RESULTS

### Microsatellite loci characteristics

After preliminary screening, a total of 35 microsatellite loci with the highest primer specificity among the 71 microsatellite loci were selected for further analysis. The number of alleles, allele size range, $H_O$, $H_E$, PIC, F(null), and HWE of the 35 microsatellite loci of polled yaks are presented in Table 1. Three monomorphic loci, namely BM2943, INRA035, and RM099, were identified. The remaining 32 loci were polymorphic. The number of alleles per locus ranged from 3 (CSSM013 and CSSM033) to 12 (SPS115), and PIC values ranged from 0.084 (MM12) to 0.815 (ILSTS028). Four of these 32 polymorphic loci had low PIC values (i.e., <0.25), 11 had moderate PIC values (0.25–0.50), and 17 loci had high PIC values (>0.50) (Table 1).

The 17 genetic markers with high PIC values were reproducible, informative, and locus-specific for parentage testing and individual identification. The number of alleles ranged from five (BM720, INRA005, TGLA227) to 12 (SPS115). Heterozygosity values ranged from 0.573 (POTCHA) to 0.806 (ILSTS028), with an average of 0.662. Deviations between $H_O$ and $H_E$ ranged from 0.003 (INRA005) to 0.103 (TGLA126). The 17 loci had an average PIC values of 0.636, ranging from 0.529 (INRA005) to 0.815 (ILSTS028). The null allele frequency ranged from 0.0015 (TGLA227) to 0.0819 (TGLA126) (Table 1).

### Hardy–Weinberg and linkage disequilibrium tests

The results of HWE tests of the 17 microsatellite loci indicated no significant differences ($P > 0.05$) (Table 1). Therefore, the 17 loci with high PIC values (>0.5) were selected for yak paternity testing. Linkage disequilibrium within polled yaks resulted in 136 comparisons, and no interlocus disequilibrium was detected between loci on the same chromosome (Table S6).

### Multiplex amplification and loading

Multiplex PCR reactions were organized to co-amplify four groups of four to five loci with high PIC values. Non-overlapping allele lengths that presented the same fluorescent color label were selected. Typical fluorescence signals of the core 17 microsatellite loci are shown in Fig. 1. Using the three available fluorescent colors, we multiplex-loaded and scored two groups of eight or nine loci in each run.

### Parentage inference

The exclusion probabilities were calculated from the allele frequencies based on the underlying assumptions of HWE. The PE1 values of the 17 core loci ranged from 0.189 (BM720) to 0.503 (ILSTS028), with an average of 0.285. The average PE2 values of the 17 markers was 0.451, and values ranged from 0.328 (INRA005) to 0.673 (ILSTS028). Regarding the 17 core loci, the combined PE1 and PE2 values were 0.99718116 and 0.99997381, respectively. For the putative parents, the combined PEP value was

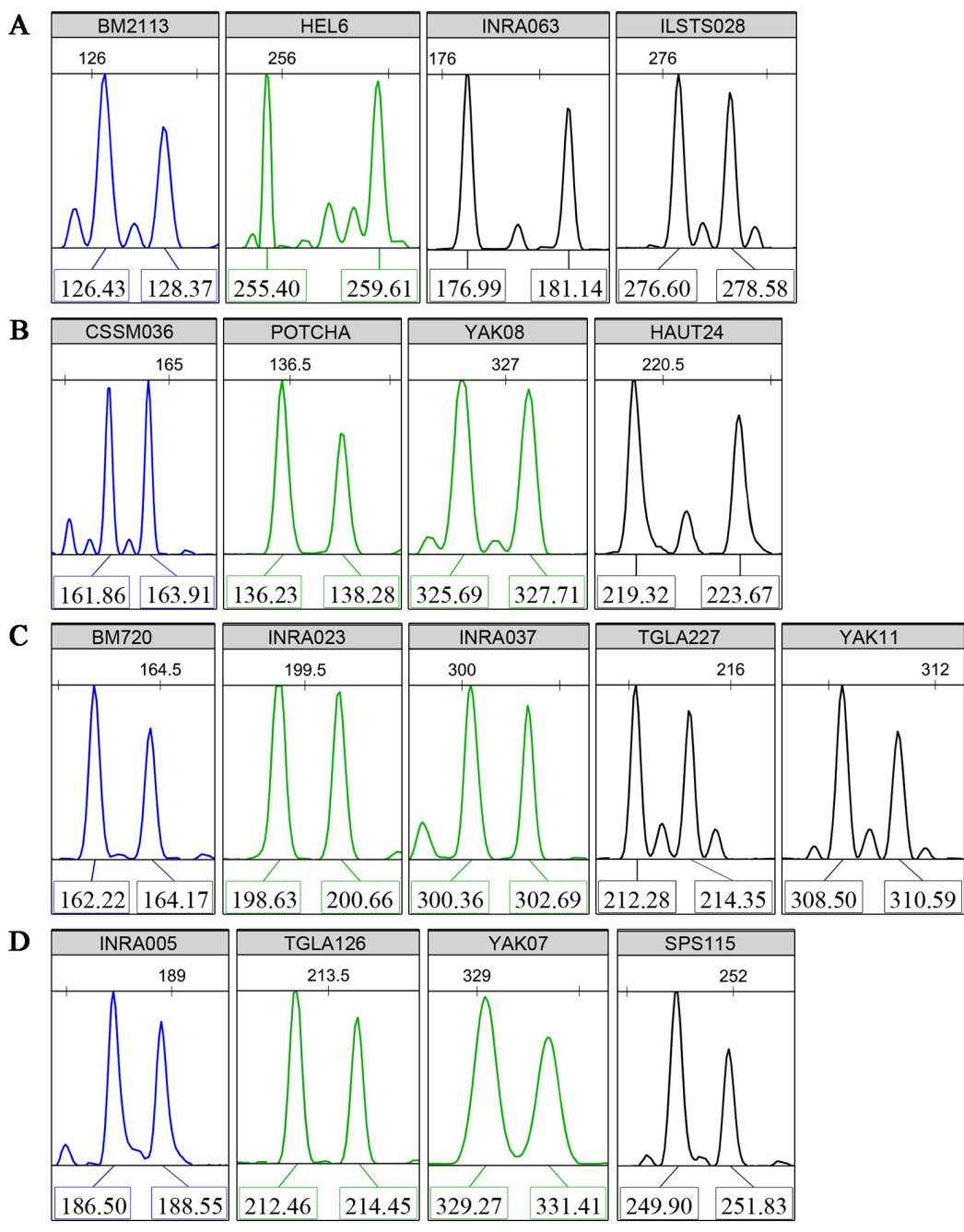

**Figure 1 Typical fluorescence signal s of detections for the core 17 microsatellite loci.** (A) Panel 1, (B) Panel 2, (C) Panel 3, (D) Panel 4. The "Panels" stand for the multiplex PCR groups. The colors of fluorochrome 6-FAM, HEX, and TAMARA are showed by blue, green, and black, respectively. The numbers above the sharp peaks represent length scales based on the internal size standard (bp). The numbers below the sharp peaks represent fragment lengths of PCR amplifications (bp).

0.99999998. Combined PEI and PESI values were >0.99999999 and 0.99999899, respectively (Table 2). Therefore, the PE values indicated that the discriminatory power of the 17 loci was high.

## DISCUSSION

Previous studies indicated that 4.3% of annual losses with regard to genetic gain during dairy breeding were caused by pedigree errors (10%), compared to simulation analysis of accurate paternity determination data (*Israel & Weller, 2000*). In fact, the pedigree error rate in yaks was high due to incorrect paternity as yaks feed primarily by grazing, thwarting parentage attribution. Additionally, clerical and insemination errors and fading ink on artificial insemination records and labels might contribute to sample mixing, thereby, leading to pedigree errors. Therefore, it is necessary to identify and correct the pedigree through parentage testing and individual identification. These practices that aim at genetic improvement of yak are essential for generating reliable breeding programs. Several reports have been published on the use of microsatellite markers for cattle identification (*Sharma et al., 2015*; *Zhao et al., 2017*), but the performance characteristics of yak identification panels have not yet been established. Parentage testing for yak breeding increases the profitability by improving the efficiency of selective breeding programs.

The most commonly used methods of livestock identification and parentage verification rely on microsatellites (*Jan & Fumagalli, 2016*; *Jeong et al., 2015*; *Wang et al., 2017*). However, single nucleotide polymorphisms (SNPs) have been applied in the identification and parentage verification of swine (*Sus scrofa*) and cattle (*B. taurus*) (*Eggen, 2012*; *Rohrer, Freking & Nonneman, 2007*). A recent study debated on the use of SNPs instead of microsatellites for parentage verification (*Kaiser et al., 2017*). However, at least 200 SNPs should be used for parentage testing to reduce false-negative results, and at least 700 SNPs are required to completely eliminate false positives (*Strucken et al., 2016*). In addition, parentage analysis based on SNPs has few predictable statistical problems that must be considered carefully and evaluated appropriately before substituting the classical STRs approach (*Amorim & Pereira, 2005*). For these reasons, microsatellites are preferred over SNPs for parentage testing.

Most of the microsatellite markers used for cattle identification and parentage verification are dimeric repeat motifs (*Carolino et al., 2009*; *Stevanovic et al., 2010*; *Van De Goor, Panneman & Van Haeringen, 2009*; *Zhao et al., 2017*). Similarly, all microsatellite markers used in the present study were dimeric. Repeat patterns have advantages as well as disadvantages. Dimeric microsatellites might have mutations or stutter bands that present allele interpretation errors (*Walsh, Fildes & Reynolds, 1996*). On the contrary, during PCR conduction, trimeric, tetrameric, and pentameric repeat motifs demonstrated lower stutter slippage efficiency than dimeric microsatellites (*Gill, Curran & Elliot, 2005*), and they ensured clear peak discrimination. On the other hand, an appropriate mutation rate might be beneficial for the verification of multigenerational parentage. This is because mutations might occur between an ancestor and a descendant's assumed father/mother, but not between an assumed father/mother and offspring, thus allowing the identification of true father/mother. For trimeric, tetrameric, pentameric, and hexameric microsatellites, gaps occurring during the sequence variant visualization within the repeat units can result in larger bin sizes than those of dimeric microsatellites (*Gill et al., 2000*). However, if the span of the sequence variants is too wide, it becomes increasingly

difficult to confine the microsatellite markers within a fluorochrome to a single multiplex PCR system. Furthermore, multimeric repeats can also be compound. All of the microsatellite markers used in the present study were dimeric containing homogeneous repeat motifs, therefore, some markers can be labeled as one fluorochrome in a single multiplex PCR with unambiguous genotyping.

In order to use them in paternity testing and individual identification, microsatellite loci must have rare null alleles, and be at HWE, and the gametic association (linkage disequilibrium) should be absent. Null alleles are not amplified to detectable levels via PCR because of mutations at primer binding sites (*Kline et al., 2011*). The frequency of null alleles is mainly estimated by Mendelian incompatibilities (*Strucken et al., 2016*) and by comparing the observed and expected number of homozygotes at a locus (*Dąbrowski et al., 2015*). This fact must be considered when performing genotyping for parentage testing and when there is an apparent opposite homozygosity between parent and offspring. In the present study, the estimated null allele frequencies remained the highest for HEL10 (F(Null) = 0.703) and MGTG4B (F(Null) = 0.163). Therefore, these loci were excluded from the core set used for parental identification.

The formulae used to estimate the exclusion probabilities assume random mating, random association between alleles of different loci, and allele frequencies consistent with HWE. The 17 loci that were selected as core microsatellite markers were at HWE (Table 1), indicating that they can be used to calculate PE values.

The microsatellites used in cattle parentage testing could not be directly applied to yak identification due to uncertainty of whether the primers used for cattle would produce the desired results in yak. In the present study, the microsatellite primers were designed based on the yak gene sequences, and were then tested by PCR and electrophoresis. In addition, the allelic frequencies differed between the cattle and yak. Even among cattle breeds, the microsatellite PIC values differed (*Mao et al., 2008*), and the same might be applied to yak breeds (*Zhang et al., 2008*). Thus, while determining the PE values of parentage testing and individual identification for a new breed, allele frequencies and PIC values should be calculated de novo.

The microsatellite loci with PIC value > 0.5 were selected for the identification panel. Nevertheless, the PIC values of these loci were still moderately lower than those used for cattle testing (*Stevanovic et al., 2010*; *Vohra et al., 2017*). Therefore, a high number of microsatellite markers were screened in the present study, and a set of 17 microsatellite markers proved to be sufficient for determining PE values. The combined exclusion probabilities of wrongly assigned sires were 99.718116%, 99.997381%, and 99.999998% for PE1, PE2, and PEP, respectively. Similar results were reported in Angeln dairy cattle, with 16 microsatellites and approximately 99.9% PE1 (*Sanders, Bennewitz & Kalm, 2006*). The PEP value calculated for the set of 17 microsatellites for parentage testing in the present study was 99.997381%, which was marginally higher than that obtained for Swiss yaks (99.5%) using 13 STR markers (*Nguyen et al., 2005*). Therefore, the set of microsatellite loci used in the present study significantly contributed to parental identification in the polled yak population.

We used several mapped cattle microsatellites to develop the sets of yak loci that were suitable for multiplex PCR amplification, and multiplex loading was conducted in a single run to reduce human errors, typing cost, and time. Nevertheless, selecting markers for a universal panel depends on the balance among the required panel accuracy, amplicon length, and ability to undergo a successful multiplex reaction. Multiplex PCR amplifications are technically more difficult than their single-locus counterparts, but they are less likely to transfer across species than single-locus amplifications. Several multiplex PCR and loading optimization methods have been investigated for parentage testing of cattle. Nevertheless, these methods cannot be directly applied to yak identification. Therefore, we developed four multiplex amplifications (each containing four or five loci) and two multiplex loads (each containing eight or nine loci) running in two gel lanes. The main advantage of this system is that the allele length ranges do not overlap within the same fluorochrome.

Hence, we proposed that a combination of 17 microsatellites can yield a polled yak panel with enhanced processing efficiency, reliability, and utility. Moreover, this system uses the standard genotyping methods of DNA fragment analysis technology. Combined with likelihood-based parentage testing, these 17 markers will help improve breeding programs and accurately determine polled yak pedigrees. If this system is used to identify polled yaks that are not the descendants of the expected breeding male yak, breeders can eliminate them from the breeding group to ensure genetic purity and breed improvement. On the other hand, if the semen samples of high-grade male yaks are mixed with those of other males, the detection system can use DNA from semen and blood of the high-grade male yak for identification. Multiplex systems can also be used to rapidly assess the history, structure, and diversity of the breeding population, and these systems can reconstruct relationships among breeds. Furthermore, these multiplex systems might also be applied to other yak breeds with gene frequencies that are similar to that of the population tested in the present study.

Although the core set of microsatellite loci presented here was meaningful for yak parentage testing, this methodology still has the following limitations: (a) Since different yak breeds have different microsatellite genotypes, this core set was only suitable for parentage testing of polled yak; (b) The PIC values of the microsatellite loci are not sufficiently high to reduce the number of loci, thus avoiding low PE values; and (c) The number of panels for multiplex PCR reactions and multiplex loading still remained too high, leading to time-consuming and high cost. Therefore, a large number of microsatellites with high PIC values should be obtained for different yak breeds to develop efficient parentage test systems, with higher PE values and fewer markers. Furthermore, microsatellites should be suitable to each specific yak breed, and markers should be screened using single multiplex PCR reactions or multiplex loading.

## CONCLUSIONS

A set of 17 microsatellite markers, which were assembled into four multiplex PCR reaction systems and genotyped in two multiplex loading systems, were identified and evaluated. The high variability displayed by these microsatellite loci demonstrated that

highly precise genotyping panels might be used for individual genotyping, parentage verification, and individual identification. The microsatellites reported in this study could also be used to evaluate yak population structure, history, and diversity, which subsequently aids the genetic improvement of domestic yak.

## ACKNOWLEDGEMENTS

We thank Hongli Jiang from Beijing UBioLab Genetics Technology Co., Ltd for technical advice on genotype evaluation and multiplex system development. We also thank Editage for improving the language of this manuscript. We thank the editor and the anonymous reviewers for their constructive comments on the manuscript.

### Funding

This work was supported by grants from the National Natural Science Foundation of China (Grant No. 31402034), the China Agriculture Research System (Grant No. CARS-37), and the Innovation Project of the Chinese Academy of Agricultural Sciences (Grant No. CAAS-ASTIP-2014-LIHPS-01). The funders had no role in study design, data collection and analysis, decision to publish, or preparation of the manuscript.

### Grant Disclosures

The following grant information was disclosed by the authors:
National Natural Science Foundation of China: 31402034.
China Agriculture Research System: CARS-37.
Innovation Project of the Chinese Academy of Agricultural Sciences: CAAS-ASTIP-2014-LIHPS-01.

### Competing Interests

The authors declare that they have no competing interests.

### Author Contributions

- Jie Pei conceived and designed the experiments, performed the experiments, analyzed the data, prepared figures and/or tables, approved the final draft.
- Pengjia Bao performed the experiments, contributed reagents/materials/analysis tools, approved the final draft.
- Min Chu performed the experiments, approved the final draft.
- Chunnian Liang analyzed the data, contributed reagents/materials/analysis tools, authored or reviewed drafts of the paper, approved the final draft.
- Xuezhi Ding analyzed the data, prepared figures and/or tables, authored or reviewed drafts of the paper, approved the final draft.
- Hongbo Wang performed the experiments, analyzed the data, approved the final draft.
- Xiaoyun Wu performed the experiments, analyzed the data, approved the final draft.
- Xian Guo conceived and designed the experiments, analyzed the data, prepared figures and/or tables, authored or reviewed drafts of the paper, approved the final draft.

- Ping Yan conceived and designed the experiments, contributed reagents/materials/
  analysis tools, authored or reviewed drafts of the paper, approved the final draft.

## Animal Ethics

The following information was supplied relating to ethical approvals (i.e., approving
body and any reference numbers):

The experimental design, sampling collection protocols, and procedures were approved
by the Animal Ethics Procedures and Guidelines of the People's Republic of China and
the Animal Administration and Ethics Committee of Lanzhou Institute of Husbandry
and Pharmaceutical Sciences of Chinese Academy of Agricultural Sciences (Permit No.
SYXK-2016-0039).

## Data Availability

The raw data are provided in the Supplemental Files.

## Supplemental Information

Supplemental information for this article can be found online at http://dx.doi.org/10.7717/
peerj.5946#supplemental-information.

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
