# Peer review of "Evaluation of 17 microsatellite markers for parentage testing and individual identification of domestic yak (Bos grunniens)"

_PeerJ, doi:10.7717/peerj.5946_

## Round 0.1 · original submission · Major Revisions

In line with Reviewer #2, I had many difficulties in discerning scientific flaws from poor English grammar. As Editor, I am inclined to assume it is the second case and therefore I give the authors the benefit of the doubt trusting that the scientific core of the manuscript is valuable. An extensive and profound revision is however required, especially in the analysis of data and statistics. In order to help the Authors to accomplish this task, I join a commented version of the manuscript with questions and suggestions, which I hope, will be useful to make it publishable. I further detail the following issues:

1. Confusion between 35 and 17 loci; discriminate in tables which ones belong to the core set.

2. Why for some polymorphic loci in Table 1, HWE was not tested?
Suggestions, doubts and some typo corrections on figures and tables:
Table 1 Genetic information on 35 polymorphic microsatellite loci
HW, departure from Hardy-Weinberg equilibrium (or change table heading to HWE); F value for null allel frequency ? is it: F- null allele frequency?

Table 2 Detailed primer information on the core 17 microsatellite loci
chromosomes. Same suggestion on statistics as in text.

Table S1 Primer information for the 71 polymorphic microsatellite loci designed according to the yak genome sequences.

Primer sequences in literature; I do not understand the content of the column headed by 'PCR product sequences' (too unrepeated; complex motif?); by 'PCR production length' it is meant 'Amplicon size'?; Location at yak genome

Table S2 Chromosomal locations of the 35 polymorphic microsatellite loci

Simplify table headings: location / accession number (Genbank?)
Table S3 Repeat number assignment for the 35 loci based on the amplification product lengths. Please add 'base pairs/bp' to Length and clarify that for each locus the 1st line corresponds to bp and the 2nd. to inferred nº of repeats.

Supplemental Table S4 Genotypes of the 35 loci in the 236 polled yaks. Please replace heading «Blood 1» with: Sample code
Supplemental Table S5 Genotype distributions of the 35 loci. Please consider reformatting the table in a more conventional/standard way, including expected values.

·

Basic reporting

No comment

Experimental design

No comment

Validity of the findings

No comment

Additional comments

I recommended minor revisions for the manuscript because the abstract is too long, I suggest that the authors reduce the description at lines 31- 38. Indeed, by the excellent quality I consider that the manuscript could be accepted in the current form.

Manuscript Peerj 22873 “Validation of 17 microsatellite markers for parentage testing and individual identification in domestic yak (Bos grunniens)”
The manuscript is clearly written in professional, unambiguous language. The supplemental files are clear and will be useful to future readers.
The most important issue is a set of 17 microsatellite markers with a high variability that could potentially be used in individual genotyping, parentage verification, and individual identification in domestic yak (Bos grunniens).
The abstract is too long, I suggest that you reduce the description at lines 31- 38 and to provide essential item for your study.
I commend the authors for this important research that could be useful for future biodiversity studies in domestic yak (Bos grunniens).

Reviewer 2 ·

Basic reporting

The study by Pei et al. proposes a state-of-the-art genotyping methodology for the problem of parentage confirmation and individual identification in yak. The study has novelty and relevance in the field of animal breeding and conservation.
Although this study has scientific merit, the manuscript suffers from poor scientific writing, and general presentation and organization. I recommend that the report is thoroughly rewritten and revised by a proficient English speaker who is also an experienced author in the field of population and forensic genetics.
A lot of attention to detail needs to be invested in this report. I am afraid that all has to be revised, from the keywords to the tables’ and figures’ legends. I am under the impression that the authors have insufficient experience in presenting scientific reports and must seek assistance to upgrade this manuscript. The vast amount of improvement that it demands at this point is beyond the scope of my responsibilities as a reviewer. However, I believe that a fresh rewrite, focus and organization can bring the study up to the standards of a scientific report for publication in an international journal such as PeerJ.
In its present form, I cannot recommend it for acceptance without major revision.

Experimental design

No comment. See above.

Validity of the findings

No comment. See above.

Additional comments

No comment. See above.

·

Basic reporting

This MS characterizes a set of 17 microsatellite loci previously used in cattle. The paper reads well and the cited literature suffices. All the most important information is supplied in the main body of the paper and as supplementary material.

Experimental design

Several families - father, mother, ofspring - as well as half-sibs should have been included in the sample panel. At least a small justification on this omission should be given.

Validity of the findings

The large sample size assures a higher accuracy of the obtained summary statistics. None of the final set showed significant frequency of null-alleles, deviations from HWE expectations or was in linkage disequilibrium.

Additional comments

This work reports a complete and almost flawless characterization of a set of microsatellites for parental testing in the yak. The minor weakness of this paper is the lack of known families and half-sibs in the sample panel. The inclusion of these samples would permit not only to verify the allele segregation as well as experimentally test the robustness of the estimated probabilities of parental exclusions.

Nonetheless, this is a clean and straightforward note that is worthy of publication.

---

## Round 0.2 · Minor Revisions

Again in line with reviewer #2, I consider the manuscript version now submitted although improved still unsuitable for publication. On my side I particularly emphasize the lack of familiarity with standards, guidelines, nomenclature, and theory of forensic genetics (although included as a keyword). As an example of serious importance is how the issue of null (silent) alleles is dealt with. Evidence for their presence can only be obtained from Mendelian inconsistencies and not from heterozygosity deficit (which can result simply from inbreeding or poor genotyping quality). That is the reason why in the cases of loci with excess of heterozygotes, negative (!) frequencies estimates are obtained.
Besides these remarks I underline the need for a thorough English language revision.

·

Basic reporting

The manuscript is written in professional, unambiguous language. The article includes sufficient introduction and background, and literature is appropriately referenced. The structure of the article is conformed to an acceptable format.

Experimental design

The investigation was conducted rigorously and to a high technical standard. Methods are described with sufficient information to be reproducible by another investigator.

Validity of the findings

The data are robust, and conclusions are appropriately stated

Additional comments

This important research providing a resource for the genetic improvement of domestic yak and could be useful for future studies of yak population structure, history, and biodiversity. I consider that the manuscript Peerj 22873 could be accepted in the current form.

Reviewer 2 ·

Basic reporting

This new version of the study has greatly improved, and the authors invested in corrections according to the reviews. I am particularly impressed by the editor's generosity in guiding the authors in this task.
However, the presentation of this work is still substandard, and further improvement is necessary to warrant acceptance for publication in PeerJ.
Again, I strongly suggest that the manuscript should be reviewed by an English-speaking geneticist.

(see comments to authors)

Experimental design

Acceptable.

Validity of the findings

Needs improvement (see comments to authors).

Additional comments

ABSTRACT

Methods
“A standardized set of microsatellite loci was assessed and applied in the parentage testing and individual identification of yak.”
This introductory phrase for the methods section of the abstract is redundant. The authors then proceed to describe the method they used, and finally say that 17 STRs were employed. Is the “standardized set of microsatellite loci” the 17 STR panel? The authors may also want to check what a standardized set of microsatellite loci means in the context of ID and parentage analyses (see e.g. van Asch et al. A new autosomal STR nineplex for canine identification and parentage testing. Electrophoresis 30(2):417-23. doi: 10.1002/elps.200800307).

Results
“These results indicated that this combination of 17 microsatellite markers could improve the efficiency, reliability, and utility of parentage testing and individual identification in yak.” Is this relative to another method that was already in use?

“Nevertheless, these loci [cattle] cannot be directly applied to yak identification because the two breeds have different genome sequences and
organization.” Cattle and yak belong to different bovid species and not breeds. Cattle are Bos Taurus and yak are Bos grunniens.

“Thus, this set of microsatellite loci are suitable for parentage/relatedness analysis in yak breeds.” This set of STRs worked for the population under study (polled yak), its use for other yak “breeds” must still be assessed.

Keywords Excessive number of keywords?


INTRODUCTION

Excessive number of references for facts that are not relevant for this particular study. Example: “The ability of yak to survive in such rugged natural environments is the result of numerous behavioral, physiological, and genetic adaptations (Barsila et al. 2014; Ding et al. 2015; Hu et al. 2012; Huang et al. 2012; Qiu et al. 2012; Wang et al. 2017b).”

“Although there are 18 yak breeds in China, only one (Datong) has been bred (Wu 2016). This phrase makes no sense to me, please clarify. Also, do the yak under study belong to the Datong breed?

“When yak graze, however, mating is very difficult to control. Semen samples could be erroneously mislabeled during preparation, and mating records might be confused because of clerical errors made during artificial insemination.” These two phrases are in contradiction. Do yak mate randomly, or is artificial insemination used?

“A highly informative set of DNA markers whose alleles can be easily scored is critical for effective paternity testing and individual identification.” This phrase is redundant.

“The present study aimed to develop and evaluate multiplex microsatellite systems that are specifically designed for polled yak parentage control and kinship analyses but that include microsatellite loci commonly used to characterize cattle genetic parameters.” Revise this phrase. According to the M&M, the authors tested mostly cattle STRs (and a few yak STRs) on polled yak, and selected an informative panel.

“The following 14 loci did not need adjustment” It’s not clear to me what adjustment means.

“Genomic DNA was extracted from white blood cells digested with proteinase K.” Did you isolate white blood cells from total blood? If not, you extracted DNA from the total blood.

“to identify repeat markers”. It’s not clear to me what this means. Do you mean that the PCR products were sequenced by Invitrogen?
“fluorescently labeled PCR products were identified by capillary electrophoresis”. Capillary electrophoresis does not identify PCR products, it separates them according to size.

“Using combinations of microsatellite markers can produce amplification fragments with high specificity, allowing a relatively cost-effective genotyping.” Unnecessary statement, everyone in the field is aware of the advantages of multiplexing.

“distinctive”: not an appropriate term in STR genotyping.

“within this subset of 17 loci.” Redundant information, it is mentioned in the previous phrase that this paragraph refers to the 17 STRs.

DISCUSSION

“On the other hand, clerical and insemination errors and stain disappearance may also influence pedigree error rates.” It is still not clear to me whether this polled yak is inseminated artificially or allowed to mate randomly. Also, it’s not clear to me what “stain disappearance”.

“the performance characteristics of a yak identification panel have not yet been distinguished”. Not clear what the authors mean by “distinguished”.

“This system uses the advancements made in DNA fragment analysis technology.” I disagree: multiplex PCR amplification with labelled primers and automatic detection of fragment size with capillary electrophoresis are standard genotyping methodology.

In my opinion, the discussion does not focus on the essential but diverges to compare STR vs SNP genotyping, the repeat structure of microsatellites (etc…), and fails to address the fundamental aspects of this work. The authors should compare their results with previous works performed on yak, and discuss the limitations of their approach (e.g. only polled yak were analysed, future developments…).


Table 1. The motif of the repeat for each loci is given (RM). How was this confirmed? The authors do not report sequencing of the alleles they have found in their sample population. If this is cattle data, it should be clarified. The authors used the yak genomic sequence to map the cattle loci, they should give the correspondent location in the yak, at least with regards to the chromosome.

---

## Round 0.3 · Minor Revisions

The improvements of this new version are still unsatisfactory and many crucial issues are still unanswered. Please consider carefully *all* the objections raised by Reviewer #2 and pay special attention to the question of loci choice. I do share with the Reviewer the objections on this matter and I add the following considerations, which may help your rebuttal. I am also not sure on what is meant by [primers used to amplify loci that] “were not adjusted for yak”. If you mean that they were not suitable – and for that you must clarify what ‘scores’ are - then a detailed explanation is required.

·

Basic reporting

Same comment to the previous review.
The manuscript is written in professional, unambiguous language. The article includes sufficient introduction and background, and literature is appropriately referenced. The structure of the article is conformed to an acceptable format.

Experimental design

Same comment to the previous review.
The investigation was conducted rigorously and to a high technical standard. Methods are described with sufficient information to be reproducible by another investigator.

Validity of the findings

Same comment to the previous review.
The data are robust, and conclusions are appropriately stated.

Additional comments

Same comment to the previous review.
This important research providing a resource for the genetic improvement of domestic yak and could be useful for future studies of yak population structure, history, and biodiversity. I consider that the manuscript Peerj 22873 could be accepted in the current form.

Reviewer 2 ·

Basic reporting

I appreciate the effort of the authors to produce a manuscript up to the standards of PeerJ. Although the manuscript is now more readable, this is not the case, and there is plenty of room for improvement. The authors have followed my suggestions, but I was disappointed to verify that they have failed to provide a deeply and self-critically revised, mature manuscript. As I stated in the comments of the previous version of this manuscript, ensuring adequate language editing and scientific writing style is beyond my responsibilities as a reviewer.
While strongly recommending that the authors use these suggestions to scan their manuscript for other opportunities for improvement, I’ll make a few comments (by no means comprehensive) as examples:

Line 41 – “The 17 loci obtained”. Loci were not obtained, they were selected from the literature and tested.
Line 42 – “Easy genotyping”. Do you mean unambiguous genotyping? “Easy” is not a scientific term.
Line 43 – “Parentage/relatedness analysis”. I have to disagree, you tested for individual identification and parentage (PE, PEI and PESI). Relatedness has a very different scope, and involves other types of kinship that you did not test for.
Line 88 – Phrase is redundant.
Line 101 – “Considered”: do you mean tested? You refer “Many studies” but you only cite two. “Relatedness identification”: please be precise on the kinship relationships that were tested in the works that you cite, if it was parentage testing and individual identification, it must be referred as so, not as relatedness.
For example, the phrase “Although many studies have considered the application of microsatellites in relatedness identification, including primary paternity testing in yak, most microsatellite loci and their primers were directly employed from a cattle reference (Li et al. 2013; Nguyen et al. 2005).” could be written as:
“Previous studies have reported microsatellite genotyping for individual identification and parentage testing in yak; however, these loci and their primers were originally developed for cattle, and used directly in yak (REFs).”
Line 99 – Eliminate “breeds”, you are improving on the transfer of cattle markers to yak, i.e. from one species to another species.
Line 101 – This paragraph needs total revision as it fails to provide a clear summary of the aim of your study. The last phrase belongs to the conclusions.
Line 113 – Replace “references” by the appropriated actual cattle references, as you did for (Li 2004).
Line 116 – What do you mean by low scores and high scores? Please be precise. There also seems to be a contraction because you say that most loci were not adapted to yak because of “low scores”, but you also say that the primers used to amplify BM1824 etc presented “high scores” and, therefore, were nor adjusted to yak. Does “adjusted” mean that you redesigned the primers based on the yak sequence?
Line 134 – It is common practice not to separate white blood cell from total blood before extraction of total DNA. Was there a reason for this separation? If yes, please explain.

Line 155 – It is not clear why you sequenced those loci, please explain. Loci were sequenced, therefore, sequences of these loci should be provided in Supplementary Material. Unidirectional sequencing may not be sufficient to survey the number of repeats and the flanking sequences correctly, due to slippage of DNAPol during the replication steps of PCR and sequencing reactions. Please discuss.

References
As I commented before, this manuscript presents a list of 49 references, which I consider excessive given the scope of this study and the depth of the discussion. Example in line 63: four references are presented supporting the significant contributions of yak to human life. I suggest that the authors decrease the number of references for introductory facts that add little to the focus of their work.

Table S1 – A column informing on the original literature refs should be included for the loci mined from previous works.

Experimental design

Please see comments on basic reporting.

Validity of the findings

Please see comments on basic reporting.

Additional comments

I appreciate the effort of the authors to produce a manuscript up to the standards of PeerJ. Although the manuscript is now more readable, this is not the case, and there is plenty of room for improvement. The authors have followed my suggestions, but I was disappointed to verify that they have failed to provide a deeply and self-critically revised, mature manuscript. As I stated in the comments of the previous version of this manuscript, ensuring adequate language editing and scientific writing style is beyond my responsibilities as a reviewer.
While strongly recommending that the authors use these suggestions to scan their manuscript for other opportunities for improvement, I’ll make a few comments (by no means comprehensive) as examples:

Line 41 – “The 17 loci obtained”. Loci were not obtained, they were selected from the literature and tested.
Line 42 – “Easy genotyping”. Do you mean unambiguous genotyping? “Easy” is not a scientific term.
Line 43 – “Parentage/relatedness analysis”. I have to disagree, you tested for individual identification and parentage (PE, PEI and PESI). Relatedness has a very different scope, and involves other types of kinship that you did not test for.
Line 88 – Phrase is redundant.
Line 101 – “Considered”: do you mean tested? You refer “Many studies” but you only cite two. “Relatedness identification”: please be precise on the kinship relationships that were tested in the works that you cite, if it was parentage testing and individual identification, it must be referred as so, not as relatedness.
For example, the phrase “Although many studies have considered the application of microsatellites in relatedness identification, including primary paternity testing in yak, most microsatellite loci and their primers were directly employed from a cattle reference (Li et al. 2013; Nguyen et al. 2005).” could be written as:
“Previous studies have reported microsatellite genotyping for individual identification and parentage testing in yak; however, these loci and their primers were originally developed for cattle, and used directly in yak (REFs).”
Line 99 – Eliminate “breeds”, you are improving on the transfer of cattle markers to yak, i.e. from one species to another species.
Line 101 – This paragraph needs total revision as it fails to provide a clear summary of the aim of your study. The last phrase belongs to the conclusions.
Line 113 – Replace “references” by the appropriated actual cattle references, as you did for (Li 2004).
Line 116 – What do you mean by low scores and high scores? Please be precise. There also seems to be a contraction because you say that most loci were not adapted to yak because of “low scores”, but you also say that the primers used to amplify BM1824 etc presented “high scores” and, therefore, were nor adjusted to yak. Does “adjusted” mean that you redesigned the primers based on the yak sequence?
Line 134 – It is common practice not to separate white blood cell from total blood before extraction of total DNA. Was there a reason for this separation? If yes, please explain.

Line 155 – It is not clear why you sequenced those loci, please explain. Loci were sequenced, therefore, sequences of these loci should be provided in Supplementary Material. Unidirectional sequencing may not be sufficient to survey the number of repeats and the flanking sequences correctly, due to slippage of DNAPol during the replication steps of PCR and sequencing reactions. Please discuss.

References
As I commented before, this manuscript presents a list of 49 references, which I consider excessive given the scope of this study and the depth of the discussion. Example in line 63: four references are presented supporting the significant contributions of yak to human life. I suggest that the authors decrease the number of references for introductory facts that add little to the focus of their work.

Table S1 – A column informing on the original literature refs should be included for the loci mined from previous works.

---

## Round 0.4 · Minor Revisions

I entirely agree with the comments from Rev#2. Please correct as suggested and perform a careful and thorough English language revision.

Reviewer 2 ·

Basic reporting

I appreciate the effort and tenacity of the authors in the revision and improvement of their manuscript. I hope that this long process has been a fruitful learning experience that will have impact on their future manuscripts, for the sake of the authors and for the sake of the reviewers. I wish the authors all the best for their future endeavors.
Most of my concerns have been addressed to my satisfaction. However, please consider the following comments.
References
In my previous review, I comment that: “As I commented before, this manuscript presents a list of 49 references, which I consider excessive given the scope of this study and the depth of the discussion. Example in line 63: four references are presented supporting the significant contributions of yak to human life. I suggest that the authors decrease the number of references for introductory facts that add little to the focus of their work.”
The authors replied: “The number of references has been reduced to three, and the number of references for introductory facts has been reduced according to the suggestion of the Reviewer.”
Despite my suggestions, the authors did not significantly decrease the number of references they use for introductory facts that do not represent important information within the scope of this work. The authors merely excluded a single reference pertaining to the specific example that I gave, and failed to revise the remaining of the manuscript.

The revision did not correct all the language mistakes. Examples:
“Of these, 17 microsatellite loci had polymorphic information content (PIC) of >0.5, and they were included in Hardy-Weinberg equilibrium without linkage disequilibrium.” Please revise.
“The 17 loci selected were deemed suitable for yak parentage testing based on their efficient amplification, unambiguous genotyping, and high PIC. Thus, this set of microsatellite loci is suitable for parentage analysis in polled yak populations.” Please revise. This second sentence is totally redundant.
“So, the study planned […]” Please revise. Studies do not “plan”, studies have aims and objectives.
“which have been commonly reported by studies on kinship relationships in cattle and yak” Please make sure that you are using “kinship” in the correct sense, and revise the manuscript with regards to this, not just the example that I am highlighting here. Parentage and kinship are different, non-interchangeable concepts. I commented on the use of the work “kinship” in previous reviews of this manuscript.
“The primers for most of the loci used in the previous studies were not adapted to the yak genome due to low scores.” Consider replacing “not adapted” with “not suitable”.
“presented relatively high scores, and therefore, were not adjusted for yak” Consider replacing “not adjusted” with “not redesigned”.
“There were no disequilibrium found in the interlocus between each locus within the same chromosome (Table S6)” Please revise this phrase, it makes no sense.
“Previous studies have indicated that 4.3% of the annual losses in genetic progress”. This phrase makes no sense, please revise. What is “genetic progress”?
I am afraid that the manuscript need to be revised again by a native English speaker with expertise in the field. I strongly suggest that JPeer editorial staff provide further assistance to the authors in the final editing.
“Genotypic disequilibrium within the polled yak resulted in 136 comparisons. There were no disequilibrium found in the interlocus between each locus within the same chromosome (Table S6).” I would rather use the most common expression “linkage desiquilibrium” instead of “genotypic disequilibrium”. The second phrase is non-sensical and must be revised.
“Therefore, the identity and parentage PEs based on the 17 core loci were highly different.” Phrase makes no sense. Do the authors mean that the PE values indicated the discriminatory power of the 17 loci was high?
These are some examples that I picked up from the revised manuscript. I suggest that the authors along with the editorial staff invest in polishing the remaining issues.

Experimental design

No comments.

Validity of the findings

No comments.

Additional comments

Please see Basic reporting.

---

## Round 0.5 · Minor Revisions

I have attached a pdf (the only format accepted) with a few remaining corrections.

---

## Round 0.6 · accepted · Accept

It seems the suggested corrections were followed.

#